# Recent Progress in Solution Processed Aluminum and co-Doped ZnO for Transparent Conductive Oxide Applications

**DOI:** 10.3390/mi14030536

**Published:** 2023-02-25

**Authors:** Mandeep Singh, Francesco Scotognella

**Affiliations:** Dipartimento di Fisica, Politecnico di Milano, Piazza Leonardo da Vinci 32, 20133 Milano, Italy

**Keywords:** transparent conducting oxides, aluminum doped zinc oxide, co-doped zinc oxide, solution processable, doping, spray pyrolysis, sol-gel, spin coating

## Abstract

With the continuous growth in the optoelectronic industry, the demand for novel and highly efficient materials is also growing. Specifically, the demand for the key component of several optoelectronic devices, i.e., transparent conducting oxides (TCOs), is receiving significant attention. The major reason behind this is the dependence of the current technology on only one material—indium tin oxide (ITO). Even though ITO still remains a highly efficient material, its high cost and the worldwide scarcity of indium creates an urgency for finding an alternative. In this regard, doped zinc oxide (ZnO), in particular, solution-processed aluminum doped ZnO (AZO), is emerging as a leading candidate to replace ITO due to its high abundant and exceptional physical/chemical properties. In this mini review, recent progress in the development of solution-processed AZO is presented. Beside the systematic review of the literature, the solution processable approaches used to synthesize AZO and the effect of aluminum doping content on the functional properties of AZO are also discussed. Moreover, the co-doping strategy (doping of aluminum with other elements) used to further improve the properties of AZO is also discussed and reviewed in this article.

## 1. Introduction

This is not an understatement to state that doped zinc oxide (ZnO) is emerging as a leading candidate for the replacement of indium tin oxide (ITO) [1,2,3]. The reason behind this claim is the excellent physical/chemical properties of ZnO, such as wide-band gap, tunable conductivity with the dopant concentration, high carrier concentration (10^20^–10^21^ cm^−3^) after doping, high transparency, ease of synthesis in different forms or morphologies, and most importantly, the high abundance of zinc on the earth crust [3,4,5,6,7,8,9,10]. In particular, its high abundance makes ZnO based devices affordable, which is one of the most important aspect of any type of technology in today’s world. On the other hand, the world-wide shortage of indium is now a well-known fact, creating an urgency for finding an alternative TCO [1]. 

Furthermore, in order to achieve optimal functional properties, ZnO was doped with many different elements, such as In, Ga, Al, Li, La, Mg, Sm, Pr, etc. [3,4,11,12]. Among them, Al- doped ZnO (AZO) exhibits the lowest resistivity (2 × 10^−4^ Ω cm), which makes it as a leading candidate to replace ITO [13], and a number of reports can be found in which doped ZnO was used for optoelectronic applications [10,14,15,16,17,18,19]. However, this low resistive film of AZO was prepared using magnetron sputtering [13], which increases its resulting cost. In this context, solution processable approaches, such as the use of sol-gel, [20] hydrothermal methods [21], etc., represents cost-effective and highly efficient methods to synthesize AZO. Beside the low-cost and easy tuning of the doping level, the solution processable route provides the possibility of synthesizing AZO in different morphologies, such as nanowires [22] nanorods [23] and nanoparticles [24], that can be beneficial for its application as a TCO. Furthermore, in addition to doping with a single element, such as Al, co-doped ZnO (doping with two different elements, e.g., Al and Ga) [25] represents an excellent approach to further enhance the functional properties of solution processed AZO for TCO applications. In fact, over the past few years, a great deal of progress have been achieved in the development of solution processed AZO and co-doped ZnO for TCO applications [12,20,22,25,26,27,28].

In this mini review, recent progress in the development of solution processed AZO and co-doped ZnO have been summarized. In the case of co-doped ZnO, only reports in which Al is used as one of the co-dopants, are included in this review article. The synthesis and thin film deposition techniques that were used to fabricate these materials are briefly discussed, while one of the most important aspects of doped-ZnO, i.e., the effect of the Al-doping level on the functional properties of ZnO, is discussed on detail. Finally, the recent literature (typically from the last 7 years) reporting on solution processed AZO and co-doped ZnO is reviewed. 

## 2. Synthesis of Al-Doped ZnO Precursor Solution and Thin Films 

Among the wet chemical or solution processable approaches, sol-gel is the most widely used method to synthesized the metal oxide nanoparticles, whether they are doped or undoped [29,30,31,32]. The reason behind its wide use is its simplicity, low cost, low operational temperature, high yield, and the possibility to synthesize complex structures and composites materials [29,30,33]. In a typical sol-gel process, the metal alkoxide precursor undergoes hydrolysis and condensation reaction, in the presence of water or alcohol as a solvent. In sol-gel, “sol” represents the colloidal particle suspension in the solvents and when these particles interconnect with each other and form a 3-dimensional network, it is called “gel” [30,34]. Moreover, when water is used as a solvent, the process is called the “aqueous sol-gel method.” On the other hand, when an organic solvent, such as ethanol, is employed, this process is termed as the “non-aqueous sol-gel method.” In the preparation of doped or undoped ZnO, most widely, zinc acetate and ethanol were employed as the precursor and solvent, respectively [4]. Coming back to the sol-gel method, the hydrolysis and condensation reactions can be expressed by the following equations,
(1)$$M-OR+H_{2}O\to MOH+ROH\;\left(Hydrolysis\right)$$
(2)$$M-OH+XO-M\to M-O-M+XOH\;\left(Condensation\right)$$
where *M* = metal and *X*, *H*, or *R* represents the alkyl group $\left(C_{n}H_{2n+1}\right)$. The detailed explanation of the sol-gel process can be found in these reports [29,30,34]. Moreover, a number of reports can be found in which AZO was synthesized by the sol-gel process in different morphologies such as nanoparticles, quantum dots, granular morphology, etc. [3,17,35,36,37,38]. 

Other solution processes that are included in this review articles are the solvothermal and hydrothermal synthesis of doped ZnO nanoparticles. Both hydrothermal and solvothermal processes can be defined as chemical reactions performed inside a closed vessel, such as an autoclave, under autogenous pressure, and the solvents are heated close to their critical points [39]. The difference between both processes is based on the type of solvent used. When water is used as a solvent, the process is hydrothermal, and when an organic solvent, such as ethanol, is used for synthesis, the process is referred to as solvothermal. However, in some cases, the term “hydrothermal” is also used when the reaction is carried out under ambient environmental conditions [39]. 

Finally, the solution obtained via sol-gel, hydrothermal, and solvothermal processes is processed on a substrates such as glass in the form of thin film using deposition techniques such as spin coating [40], spray pyrolysis, [41] dip-coating, [42] etc. Furthermore, in addition to these deposition techniques, aerosol-assisted CVD (AACVD) [25], electrospinning [43], and electrodeposition [23] are also considered because the starting material is in solution form (precursor solutions). 

The detailed discussion of these thin film depositions and nanostructure synthesis techniques is out of the scope of this mini review article. Readers can refer to these articles for details [44,45,46,47,48,49]. 

## 3. Finding an Optimal Doping Level of Al: Effect of Al Doping on Functional Properties of AZO

The doping of ZnO with aluminum (Al) offers several advantages, such as high transparency, excellent conductivity, non-toxicity, and high mechanical/chemical stability that make this material ideal for TCO applications [7,21]. However, the doping concentration of Al needs to be carefully selected or optimized, as it can also decrease the electrical, structural, and optical properties of the AZO [50,51,52,53]. In this section, we will discuss a few reports in which the important factors associated with the Al doping level that can deteriorate the properties of AZO are presented. 

Y. Zhao et al. [54] have studied the influence of Al doping on the optical and electrical properties of Al*_x_*Zn_1−*x*_O (for *x* = 0.0625, 0.125, or 0.1875) via first-principal investigations based on the density functional theory (DFT). Detailed theoretical observations reveal that the band gap and electrical conductivity of AZO were found to be decreased with the increase in ‘x,’ i.e., the doping level of Al. This can be understood from the dependency of the AZO band structure on the Al doping level, as shown in Figure 1. The calculated band gap of pure ZnO was found to consist of three parts: the upper part (−3.8 to 0 eV), the lower part (−6.7 to −3.8 eV), and the third part, which resulted from the interactions between lower and upper parts, and which lies between −18.6 to −16.0 eV. The band structures of AZO at different doping levels of Al are presented in Figure 1b–d. Clearly, doping with Al caused the decrease in the energy levels of both the conduction and valance bands, which resulted in the shift in the Fermi level toward the conduction band (the Fermi level becomes located in the conduction band). According to the theory of semiconductor when the Fermi level lies inside the conduction band, the quantum states near the conduction band minimum become completely filled with electrons. This distribution of electrons obeys the Fermi distribution law, but disobey the Boltzmann distribution conditions. When the doping level of Al continuously increased, the interactions between the impurities also increased, and the band-gap narrowed. In fact, the bulk doping of ZnO with Al generates the additional potential field around the Al due to its ionization, causing the scattering of electrons that results in the decrease in electrical conductivity. 

Furthermore, W. Sripianem investigated the effect of Al doping levels, as a function of the dissolution of the aluminum precursor in a solvent, on the properties of AZO thin films [51]. The crystallinity of thin film was found to decrease with the increase in dopant concentration, owing to the decrease in the grain size, as shown in Figure 2. In particular, all doped thin films exhibit high transmittance (>80%), and electrical conductivity was improved only when ZnO was doped with 1 at% and 2 at% aluminum. The author suggests that at a higher doping level (>2 at%), the conductivity of the AZO film decreased because of the limited solubility of the aluminum precursor in the solvent. Additionally, the occurrence of high grain boundary electron scattering due to the smaller grain size also lowers the conductivity at higher doping levels (see Figure 1). These findings indicate that the dissolution of aluminum precursors in a solvent is an important factor that should be consider before AZO synthesis.

In another work, Al-doped ZnO thin films with different dopant percentage (1 at.%, 3 at.%, and 5 at.%) were prepared by the dip coating method on a glass substrate [42]. The precursor solution (zinc acetate dihydrate) was prepared by the sol-gel method. A detailed investigation reveals the formation of the polycrystalline AZO thin film with the highest transparency when 1% doping of aluminum was performed. In particular, AZO films exhibit minimum and maximum absorption for 1% and 5% Al doping. The band gap of AZO films continues to increase until reaching a 3% doping level and decreased at a 5% doping level (due to the red-shift in the absorption spectra of AZO at 5% doping). The author suggested that this red-shift is the indication of a stress relaxation mechanism, which is due to the merging of the impurity level into the conduction band of AZO (also causing the decrease in the band gap). 

Furthermore, excellent insight was presented by Jianwei Li. et al. [52] on variation in the electrical properties of AZO with Al doping content, as shown in Figure 3. The AZO films were prepared by aerosol-assisted chemical vapor deposition (AACVD). Clearly, both mobility and resistivity were abruptly decreased when 2.9 at% Al doping was performed, while carrier concentration at this doping level was abruptly increased. The author suggests that the abrupt increase in carrier concentrations at 2.9 at% Al doping is due to the release of one electron for every Zn^2+^ substituted by Al^3+^. On the other hand, a decrease in mobility was observed due to the ionized impurity scattering. In fact, at the 2.9 at% doping level, the AZO films exhibit the lowest resistivity, i.e., 3.54 × 10^−3^ Ω cm. 

In a similar work, the optimal structural, electrical, and optical properties of AZO spray coated film was observed at 1 at% of Al doping [50]. When doping was increased beyond this value, both the crystallinity (see Figure 4) and conductivity of the AZO films were found to decreased. Moreover, the Hall effect measurements showed the change in carrier concentration, carrier mobility, and resistivity upon doping of ZnO with Al. It has been suggested that the difference in an ionic size of Zn (r_Zn_^2+^ = 0.074 nm) and Al ((r_Al_^3+^ = 0.074 nm) results in the occurrence of stress in AZO film at higher doping levels. This stress deteriorates the structural properties of the AZO films. As far as the decrease in conductivity of the AZO film is concern, many reports also indicate that a higher Al doping level caused the formation of a non-conducting Al_2_O_3_ phase. This non-conducting phase creates disorder in the crystal structure and also acts as a carrier trap [55,56].

Many other reports can be found in which the effect of the Al doping level on the functional properties of AZO are described [53,57,58,59].

In summary, the excess doping of aluminum in ZnO and the limited dissolution of the aluminum precursor in a solvent at a higher doping content can harm the properties of AZO. Indeed, all the articles discussed above suggested different optimal doping levels of Al, thus making it difficult to pin-point one doping level that can be adopted to synthesize highly efficient AZO. Hence, finding an optimal Al doping level in ZnO that can offer adequate electrical, structural, and optical characteristics for use as a TCO is a challenging and complex task.

## 4. Aluminum and co-Doped ZnO 

In this section, we discuss some of the interesting reports on AZO and co-doped ZnO published over the last 7 years. It should be noted that in co-doped ZnO, only reports in which Al was included as one of the co-dopants were considered. Furthermore, while discussing these reports, attention has been paid to the synthesis approach, the thin film deposition techniques, and the effect of parameters such as doping content, precursor type, thin film annealing temperature, solvent type, etc. on the functional properties of doped ZnO.

### 4.1. Aluminum Doped ZnO (AZO)

A great deal of progress has been made over the last few years in the development of AZO for TCO applications. Q. Nian et al. [60] tuned the electrical and optical properties of sol-gel spin-coated AZO thin film via ultra-violet laser crystallization (UVLC). Specifically, UVLS treated films exhibit improved high transmittance (88–96%) and electrical resistivity of 1 × 10^−3^ Ω cm. Moreover, AZO film exhibited extremely low scattering transmittance (1.8%), which was found to be superior compared to that of solution deposited silver nanowires. Furthermore, AZO nanofibers were prepared by the electrospinning technique, while the precursor solution was prepared by the sol-gel method [43]. The author suggested that the nanofiber transparency increased when annealed in air. This is an important development in the synthesis of AZO, as one-dimensional (1D) nanostructures, such as nanofibers, nanowires [61], etc., offer excellent electrical properties, along with high crystallinity, that can be beneficial for optoelectronic application. However, as far as this work on nanofibers is concerned, electrical properties were not investigated by the authors. Furthermore, T. Ganesh et al. [62] synthesized a multilayer (6–24 layers) thin film of AZO (Al 1.5 wt%) via the sol-gel spin coating technique. In up to 18 layers, the crystallinity, crystallite, and grain size were found to be improved, while all the films maintained more than 85% transparency. Moreover, AZO thin films with layers between 10–18 possessed low-activation energies, a better dark-photo current, and good photo response, making them ideal for solar cell applications. Furthermore, a low-temperature spray-coated AZO thin film, with varying concentration of Al (0.25 at.%, 0.50 at.%, 0.75 at.%, and 1at.%) was reported for use in transparent electronics [63]. The schematics of the spray pyrolysis system and the AZO thin film growth process, respectively, are shown in Figure 5a,b. In particular, at the optimal doping level (0.5 at.%), the AZO thin films exhibit low resistivity (4 × 10^−3^ Ω cm), high crystallinity, and 87% optical transparency. 

Furthermore, not only the doping level, but also the properties of the solvents used to prepare the precursor solution were found to affect the properties of the AZO film. D. B. Potter et al. [64] used many different solvents (as well as a mixture of two solvents), i.e., methanol, n-hexane, toluene, tetrahydrofuran, cyclohexane, and ethyl acetate to prepare the precursor solution for AZO. The films were deposited via the AACVD technique. It has been observed that the films prepared using methanol (MeOH) exhibit high transparency (83%) (see Figure 6), low-resistivity (0.5 × 10^−2^ Ω cm), and an optical band gap of 3.25 eV owing to the lowest boiling point of MeOH as compared to those of the other solvents. Due to this, MeOH was more likely to evaporate during the growth process of the AZO film via AACVD as compared to other solvents, which in-turn affected the properties of AZO. Furthermore, post-annealing treatment was found to enhance the electrical properties of the AZO thin films (2 at% Al) deposited by spray pyrolysis [26]. In particular, upon annealing at 450 °C under vacuum, the resistivity of the film was found to be improved from 1.39 × 10^−2^ Ω cm (as deposited) to 5.10 × 10^−3^ Ω cm (annealed). However, the transmittance of the AZO film slightly decreased after annealing, but it still exhibited a transmittance of higher than 85%. This slight decrease in transmittance was attributed to the increase in phonon scattering and the free charge carrier absorption of photons, as annealing under vacuum caused the increase in the free charge carrier concentration and mobility. 

Furthermore, AZO films (see Figure 7) with different doping contents of Al were fabricated using fine-channel mist chemical vapor deposition (FCM-CVD) [65]. The precursor solution was synthesized via dissolving zinc chloride (ZnCl_2_) and aluminum chloride hexahydrate (AlCl_3_·6H_2_O) in distilled water. Detailed observations reveal that with an increase in the doping concentration, the preferential orientation of ZnO changes from (002) to (100). Moreover, the AZO films with doping concentrations of 25% and 50% were found to exhibit high transmittance, making them ideal for TCO applications. 

Furthermore, it is well known that AZO film prepared using vacuum deposition techniques, such as sputtering, exhibit properties superior to those of solution processed films [28]. Especially, solution processed AZO films exhibit high electrical resistivity [66]. To tackle this issue, A. Kumar et al. [66] suggested that vacuum/low pressure annealing is a viable route to enhance the electrical properties of solution processed AZO film. In this work, AZO films were fabricated using spray pyrolysis and after vacuum annealing, films showed an enhanced transparency of 82%, as well as low resistivity (2 × 10^−3^ Ω cm) as compared to the as deposited films. 

In the Table 1, some of the interesting reports regarding solution processed AZO in different forms, such as thin films, nanofibers, nanowires, etc., are listed, along with their functional properties.

### 4.2. co-Doped ZnO (Al Doping with Other Elements) 

Co-doping is another interesting approach to tune the functional properties of ZnO for TCO applications. In fact, previously we have seen that after a certain doping level, the functional properties of AZO, especially its electrical conductivity, are decreased, either due to the insolubility of the aluminum precursor, or by excess aluminum doping [51,52,54]. For practical application, AZO must possess low resistivity, and in this context, the co-doping of AZO with another element offers a pathway to further improve the properties of AZO [4]. Moreover, an ideal TCO must possess not only good electrical conductivity, but also high transparency in the visible region and adequate structural properties. However, as specified in many reports discussed in the previous section, Al doping caused the improvement in one or two properties, but other properties required for its application as a TCO were compromised. For example, as shown in Figure 3, with an increase in Al doping, the carrier concentration is increased and resistivity is decreased, but the mobility of AZO is also decreased. The mobility of TCO is also an important parameter, as it determines the charge transfer from TCO to the active layer. Hence, co-doping can be beneficial to achieve the optimal functional properties of AZO. 

In an interesting work involving an AIZO (Al and In co-doped ZnO) thin film, the effect of the precursor type and the doping level of aluminum and indium on the functional properties were investigated [71]. The films were prepared by ultrasonic spray pyrolysis. Detailed investigations reveal that by using aluminum chloride as a precursor and a doping level of 2 at.% of Al and In, the AIZO film possesses superior structural, optical, and electrical properties. It has been revealed that the better decomposition of the aluminum chloride precursor than aluminum sulphate plays a key role in determining the properties of AIZO films. In another work, the zinc precursor was ball milled before the preparation of the co-doped (1.5 at% Al and 1.5 at% In) AIZO precursor solution using the sol-gel method [72], while the AIZO thin film was prepared by ultrasonic spray pyrolysis. The film exhibits high crystallinity along (002), >70% transparency, and low resistivity (2.35–4.59 × 10^−3^ Ω cm). Furthermore, Zi N. Ng et al. [73] fabricated the Al-Ga co-doped ZnO microrods via the sol-gel spin coating technique. Indeed, the microrod thin film exhibited high transparency (95%), with lowest resistivity, determined by Hall effect, of 23 Ω cm when the doping level was fixed at 1 at%. 

Furthermore, D. B. Potter et al. [25] synthesize three different types of thin films, i.e., aluminum/gallium co-doped ZnO (AGZO), indium/gallium co-doped ZnO (IGZO), and aluminum/indium co-doped ZnO (AIZO) thin films via aerosol assisted chemical vapor deposition (AACVD). The AGZO and AIZO films were found to exhibit similar morphology, with randomly oriented grains (see Figure 8 (left)), while the grains of the IGZO film were more likely hexagonal in shape. The optical characterization results, as shown in Figure 8 (right), represent the high transmittance of the AGZO film as compared to the others. The reason behind the low transmittance (high absorption) of both IGZO and AIZO is the larger radius of the In^3+^ ions that creates disorder in the structure and increases the optical absorption. In addition to better optical properties, AGZO films also possess superior transport properties, such as the lowest resistivity, the highest carrier concentration, and carrier mobility, making this thin film suitable for TCO applications. 

Furthermore, the rare earth element samarium (Sm)-doped AZO (Sm:AZO) was deposited via the nebulizer spray pyrolysis technique [12]. The precursor solution was prepared via the solution route, and the aluminum doping level was fixed at 3 at%, while different doping levels of Sm were employed (0 at%, 0.5 at%, 1 at%, and 1.5 at%). Indeed, with an Sm doping level of 1 at%, the co-doped AZO films exhibited high transparency (90%), with an approximate energy gap of 3.30 eV (see Figure 9). Moreover, Sm:AZO films exhibited low electrical resistivity, i.e., 4.31 × 10^−4^ Ω cm. In fact, the doping of ZnO with this rare earth element offered unique opto-electronic properties due to their highly discrete energy levels [4]. 

In particular, doping with rare earth elements found to enhance the electrical and optical properties of ZnO, which can be beneficial as a TCO [4]. In this regard, praseodymium (Pr) co-doped AZO thin film was prepared by the nebulized spray technique [11]. Again, the Al doping content was fixed at 3 wt%, while different doping concentrations of Pr were adopted. In Figure 10, atomic force microscopic images of the Pr:AZO films at different Al doping levels are shown. The doping with Pr was found to enlarge the size of the spherical grains as the doping level increased. In particular, the grain sizes were found to be 45 nm, 56 nm, 62 nm, and 68 nm for doping levels of 0%, 0.5%, 1%, and 1.5%, respectively. Furthermore, with the 1.5% doping, the Pr:AZO films exhibited a low resistivity of 4.62 × 10^−4^ Ω cm and high carrier concentrations of ^−3^; thus, they were proposed to be ideal for optoelectronic applications. Furthermore, Lee et al. [74] synthesized the Sn and Al co-doped ZnO (TAZO) thin film using the sol-gel dip coating process and investigated the resulting structural, electrical, and optical properties. Figure 11 showed the cross-sectional SEM images of TAZO film at different doping levels. The thickness of all the films was found to be 350 nm. The films were found to be highly transparent, with an average transmittance of 88%, while the film with the doping level fixed at 1at% for both the elements exhibited the minimum resistivity (0.36 Ω cm). 

In another work, F. Khan et al. [75] investigated the charge transport properties of solution processed Ag and the Al co-doped ZnO nanostructures. To synthesize the nanostructures, the Al/Zn molar ratio was fixed at 0.5%, while the Ag/Zn molar ratio varied to different values such as 0, 0.3, 0.5, and 1%. In Figure 12, variations in resistivity, carrier density, and mobility were expressed as a function of the Ag/Zn molar ratio (R_Ag/Zn_). Cleary, the nanostructures exhibited better charge transport properties, including low resistivity, high mobility, and carrier concentrations at the 0.3% molar ratio. Moreover, the nanostructures also exhibited high transparency (85%) and an optical band gap of 3.14 eV. 

Furthermore, the In and Al co-doped ZnO (AIZO) thin films were prepared with spray pyrolysis, and the precursor solution was synthesized via the sol-gel process [41]. While investigating the effect of different Al and In doping levels on the properties of the AIZO thin film, the author suggested that 1.5% is the optimal level for In/Al. Indeed, at this optimal doping level, the AIZO films exhibited a low electrical resistivity of 3 Ω cm and a high transmittance of 85%. In Table 2, different co-doped ZnO, with their functional properties, are summarized.

Hence, co-doping is an alternative approach to further optimize the functional properties of solution-processed AZO. 

## 5. Conclusions

The recent progress in the development of solution-processed AZO and co-doped ZnO is systematically reviewed in this article. According to the literature, sol-gel is found to be the preferable chemical route to synthesize the precursor solution of AZO and co-doped ZnO, while spin-coating and spray pyrolysis are most commonly employed to deposit their thin films. The AZO film deposited via spray pyrolysis, with an aluminum doping level of 2 at%, was found to exhibit the lowest resistivity, i.e., 5.10 × 10^−3^ Ω cm. Furthermore, besides the improvement in AZO properties, the excess doping of Al and the limited solubility of the Al precursor can also harm the properties of AZO. Thus, the AL doping level must be carefully selected and optimized. In this context, co-doping represents an interesting route to further enhance the functional properties of AZO. Among the various co-dopants, praseodymium (Pr) doped AZO exhibited the lowest resistivity, i.e., 4.62 × 10^−4^ Ω cm. Clearly, by choosing the appropriate co-dopant, along with aluminum, co-doped ZnO exhibited better properties, especially in regards to low resistivity, than AZO. Furthermore, beside the aluminum doping level, the type of solvent and precursor are also found to affect the properties of AZO. Specifically, solvent with a low-boiling point (e.g., methanol and ethanol) tends to evaporate rapidly during the deposition of the film, leading to the improved functional properties of AZO as compared to those of the solvents with a higher boiling point (e.g., n-hexane, toluene, etc.), while the aluminum precursor (e.g., aluminum chloride) with better decomposition also determined the final properties of AZO.

Hence, solution-processed AZO and co-doped ZnO are excellent candidates to replace ITO, provided that some of the important parameters, such as the aluminum doping level, the type of solvent, and the precursor, are carefully chosen. Moreover, in addition to its granular structures, AZO should also be synthesized and explored in other morphologies, such as quantum dots, nanowires, nanoparticles, etc., to achieve this goal. 

## Figures and Tables

**Figure 1 micromachines-14-00536-f001:**
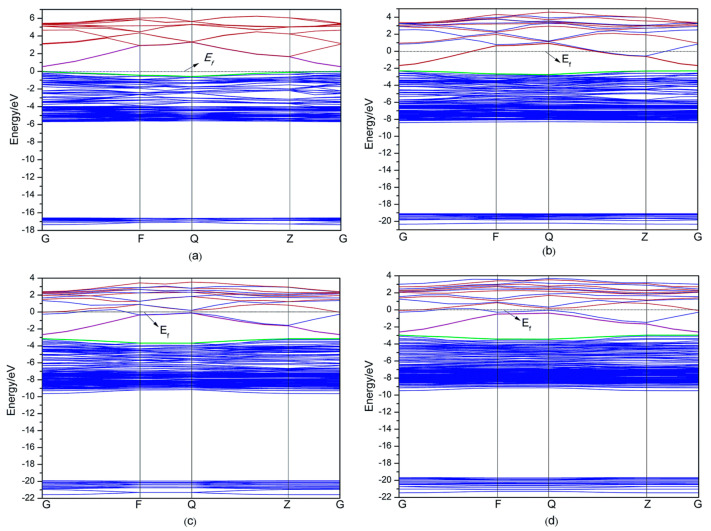
Band structures of Al_*x*_Zn_1−*x*_O with various amounts of doped Al: (**a**) *x* = 0, (**b**) *x* = 0.0625, (**c**) *x* = 0.125, (**d**) *x* = 0.1875. Reproduced with permission from ref. [54].

**Figure 2 micromachines-14-00536-f002:**
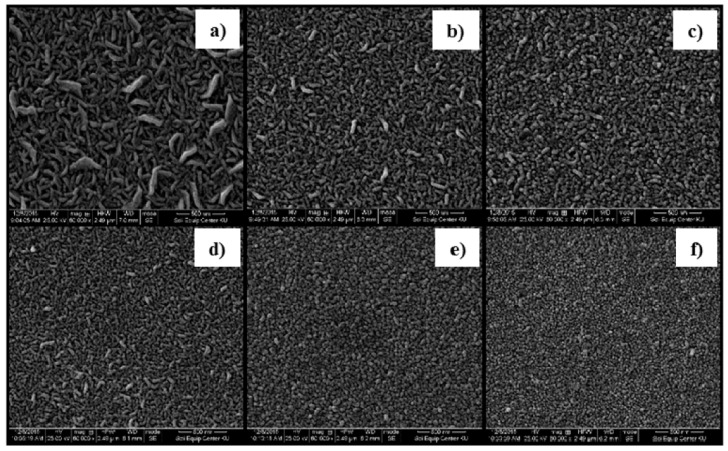
SEM images for AZO films obtained from spray pyrolysis with varying Al-doping contents: (**a**) 0 at.% (undoped), (**b**) 1 at.%, (**c**) 2 at.%, (**d**) 3 at.%, (**e**) 4 at.%, and (**f**) 5 at.%. Reproduced with permission from ref. [51].

**Figure 3 micromachines-14-00536-f003:**
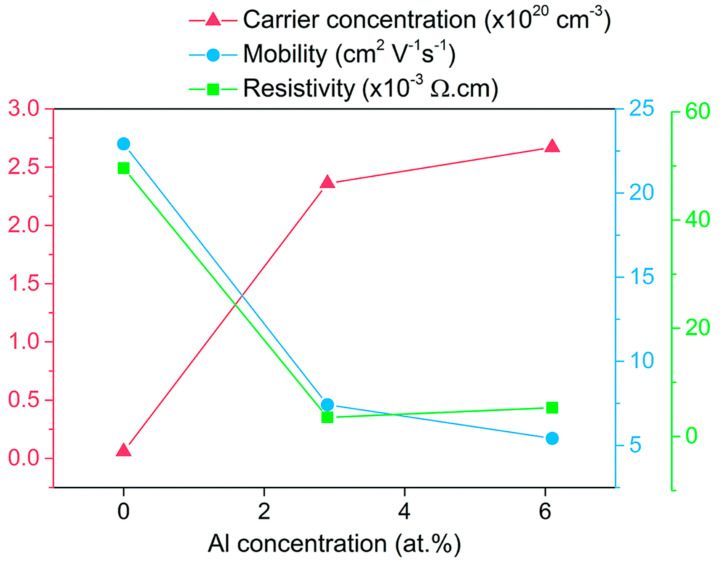
Hall effect results showing the change in carrier concentration, carrier mobility, and resistivity upon doping of ZnO with Al. Reproduced with permission from ref. [52].

**Figure 4 micromachines-14-00536-f004:**
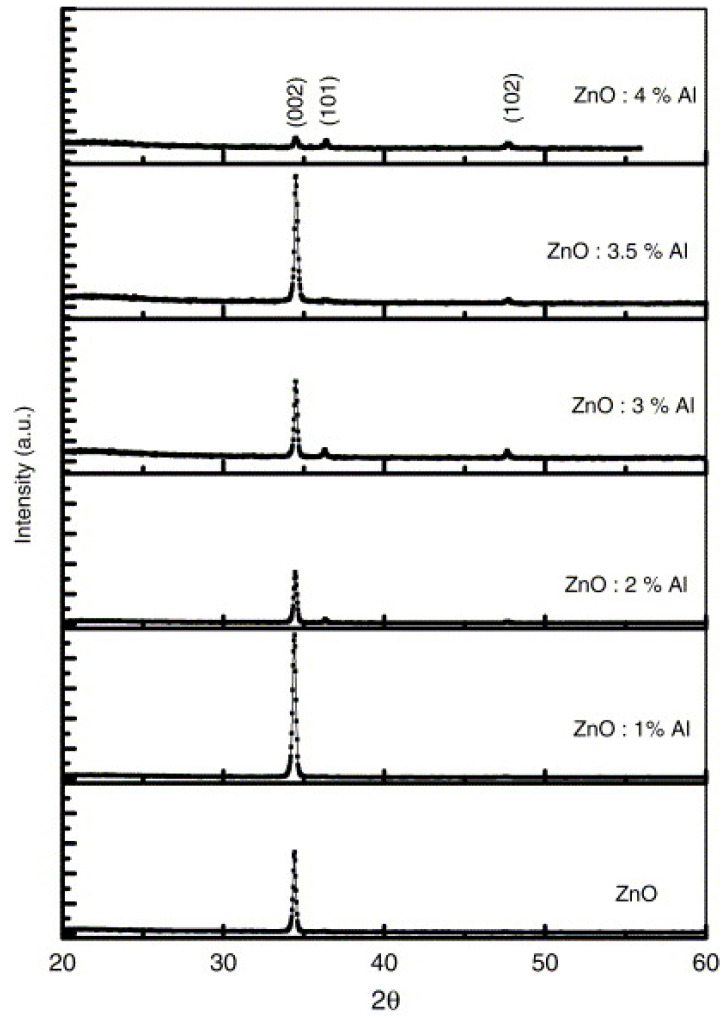
XRD spectra of undoped ZnO and AZO films deposited at 400 °C on quartz substrates for various Al doping concentrations. Reproduced with permission from ref. [50].

**Figure 5 micromachines-14-00536-f005:**
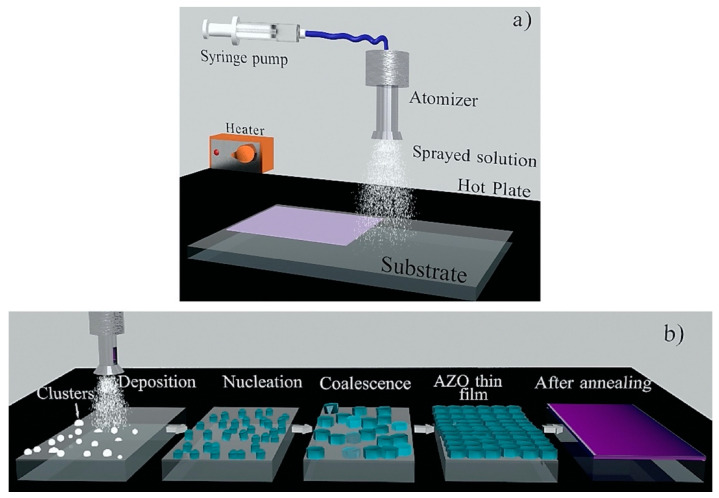
(**a**) Simplified schematic of the spray pyrolysis system and the main stages (**b**) of the ZnO-based film growth process, starting from small sprayed clusters, which undergo nucleation and coalescence, forming a continuous polycrystalline layer that then acquires a light-purplish color during substrate heating (via the hot plate), and finally changes to a darker purple after the rapid thermal annealing (RTA) post-process. Reproduced with permission from ref. [63].

**Figure 6 micromachines-14-00536-f006:**
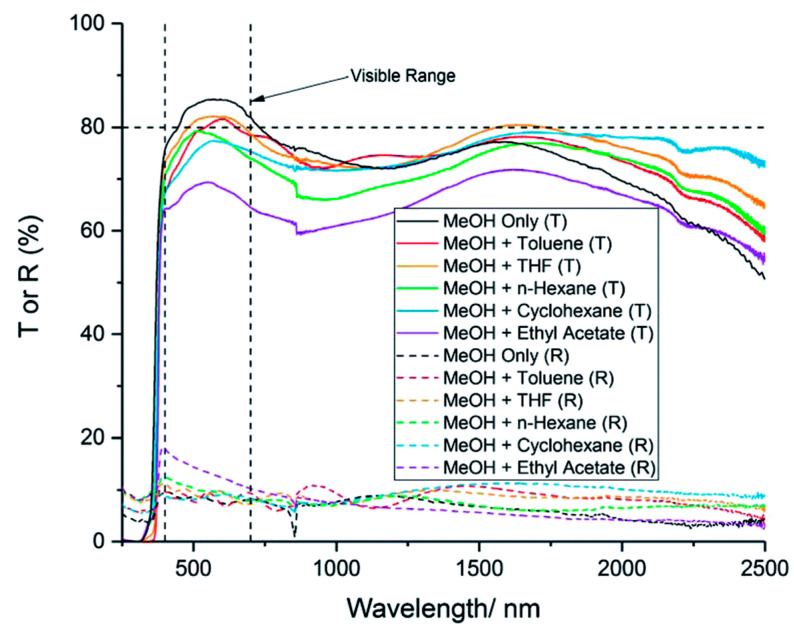
Transmission–reflectance spectra for 10 mol% AZO deposited via AACVD using different solvents. “T or R” refer to transmission or reflectance. Reproduced with permission from ref. [64].

**Figure 7 micromachines-14-00536-f007:**
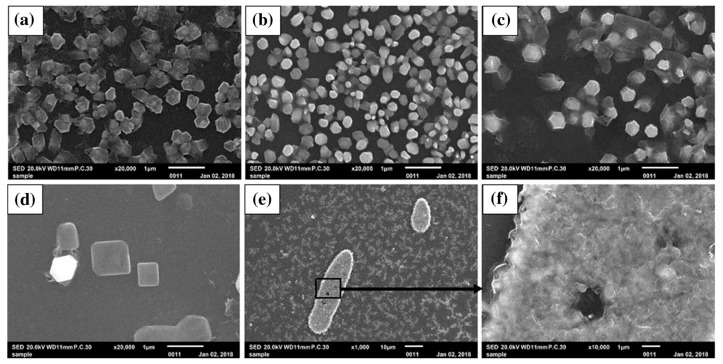
Surface images of undoped and doped ZnO films, (**a**) undoped, (**b**) 5% Al doping, (**c**) 10% Al doping, (**d**) 25% Al doping, (**e**) 50% Al doping, and (**f**) zoomed-in image of the zone indicated by the black rectangular box depicted in (**e**). Reproduced with permission from ref. [65].

**Figure 8 micromachines-14-00536-f008:**
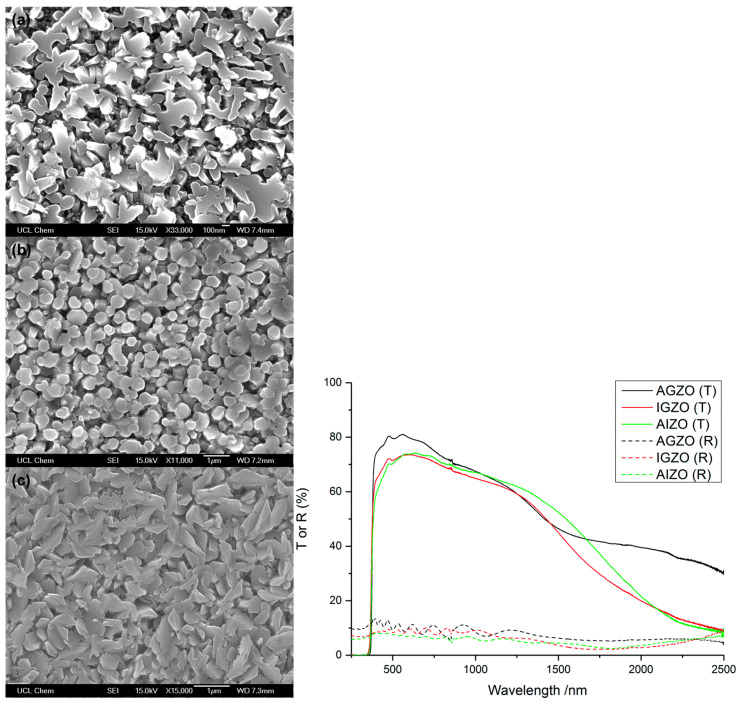
(**Left**) SEM images of the (**a**) AGZO, (**b**) IGZO, and (**c**) AIZO thin films. (**Right**) Transmission–reflectance spectra of the AGZO, IGZO, and AIZO films. All the films were deposited at 450 °C. Reproduced with permission from ref. [25].

**Figure 9 micromachines-14-00536-f009:**
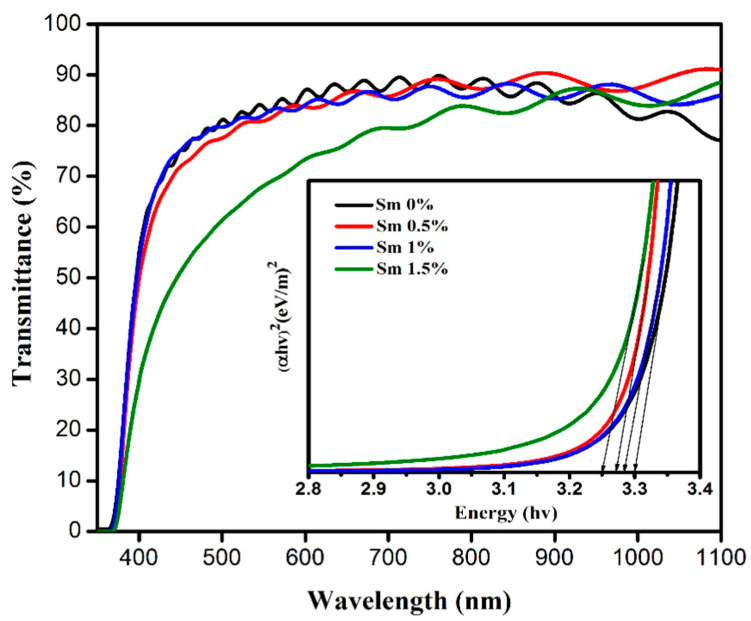
Optical transmittance of Sm:AZO thin films with different co-doping levels (inset of figure), showing the relationship of (αhυ) 2 and hυ of Sm:AZO thin films with different co-doping levels. Reproduced with permission from ref. [12].

**Figure 10 micromachines-14-00536-f010:**
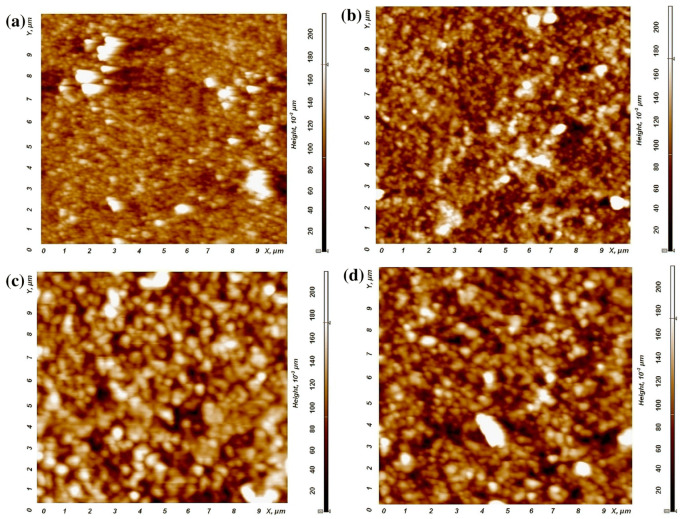
The 2D AFM images of Pr:AZO thin films with different doping levels: (**a**) 0%, (**b**) 0.5%, (**c**) 1%, and (**d**) 1.5%. Reproduced with permission from ref. [11].

**Figure 11 micromachines-14-00536-f011:**
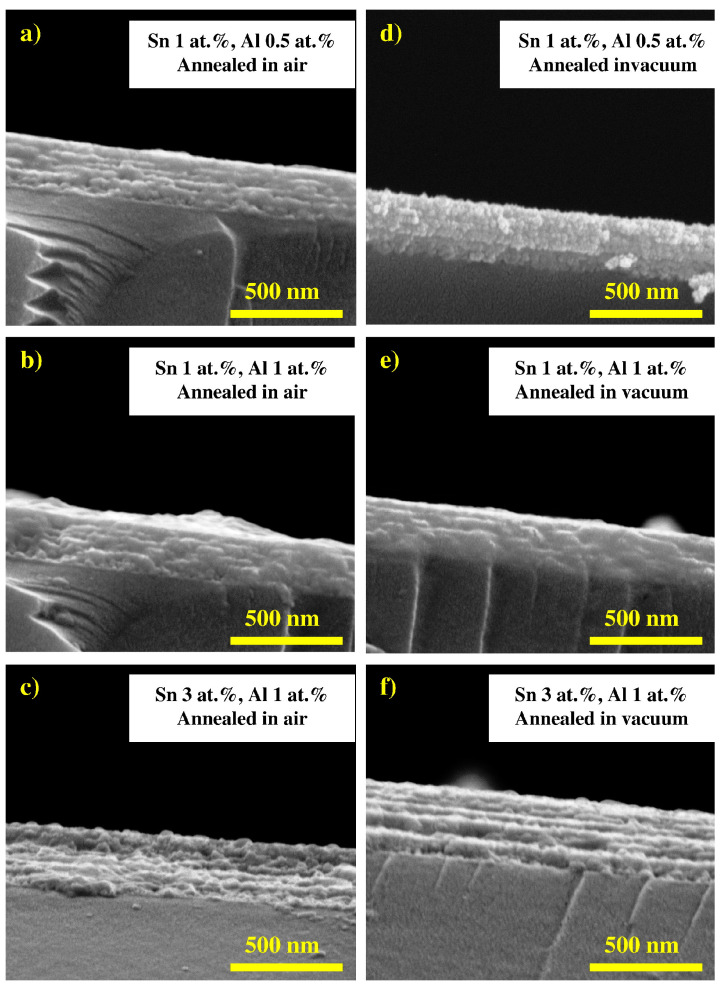
The cross-sectional SEM image of the TAZO thin film, with varying dopant concentrations, annealed at 500 °C for 1 h in air (**a**–**c**) and vacuum (**d**–**f**). The doping levels are (**a**,**d**) Sn = 1 at.% and Al = 0.5 at.%; (**b**,**e**) Sn = 1 at.% and Al = 1 at.%; and (**c**,**f**) Sn = 3 at.% and Al = 1 at.%. Reproduced with permission from ref. [74].

**Figure 12 micromachines-14-00536-f012:**
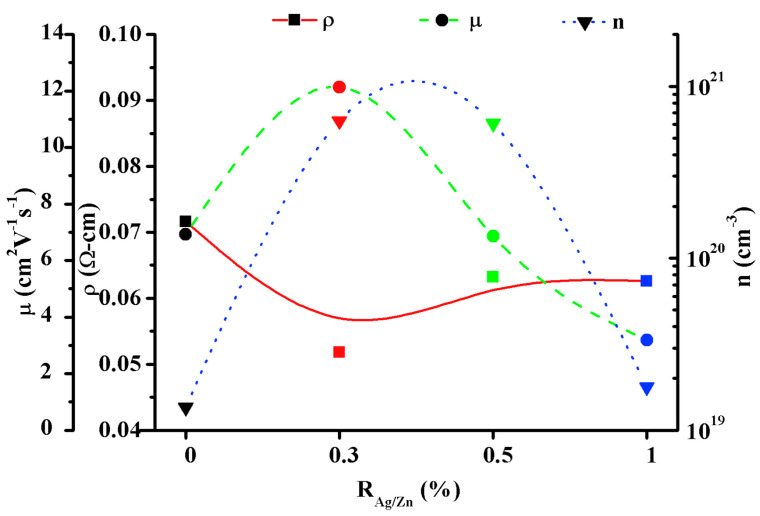
Variation in resistivity, charge carrier density, and mobility of the ZnO:Al:Ag nanostructures with Ag doping. Reproduced with permission from ref. [75].

**Table 1 micromachines-14-00536-t001:** Solution processed AZO in different forms, such as granular, nanofibers, nanorods, etc., along with their functional properties. Here, ODL = optimal doping level (at%) of aluminum; T = transmittance (%) of AZO in the visible region; E_g_ = band gap (eV) of AZO; and *ρ* = resistivity (Ω cm) of AZO.

AZO	Process	Morphology	ODL (at%)	T (%)	E_g_ (eV)	*ρ* (Ω cm)	Ref.
Thin film	Sol-gel, spin coating	Nanocrystals	2	88–96		1 × 10^−3^	[60]
Thin film	Sol-gel, spin coating	Grains	1	80–95		50	[20]
Nanofiber	Sol-gel, electrospinning	Nanofibers	1	90		-	[43]
Thin film	Sol-gel, ultrasonic spray pyrolysis	Grains	8	80	3.27	9	[67]
Thin film	Sol-gel, dip coating	Nanoparticles	3	99	3.25–3.32	-	[24]
Thin film	Sol-gel, ultrasonic spray pyrolysis	Grains	3	>85	3.3	-	[27]
Thin film	sol-gel, dip coating	Grains	1.4	>90	3.29	2.1 × 10^−2^	[68]
Thin film	Sol-gel, spin coating	Grains	2	>75	3.25–3.30	-	[69]
Thin film	Sol-gel, ultrasonic spray pyrolysis	Grains	0.5	87	3.23–3.35	4 × 10^−3^	[63]
Thin film	Aerosol-assisted CVD (AACVD)	Grains	2.9	84	3.40	3.54 × 10^−3^	[52]
Particles	Solvothermal method	Particles	1–9	~80	2.84–3.36	-	[70]
Thin film	Sol-gel, spin-coating	Nanostructures	2	97	3.39–3.37		[40]
Nanowire	Hydrothermal method	Nanowires	1–3	>80	3.23–3.37	-	[22]
Nanorods	Electrodeposition	Nanorods	1–2	61–82	-	-	[23]

**Table 2 micromachines-14-00536-t002:** Solution processed co-doped ZnO in different morphologies, along with their functional properties. Here, AIZO = Al, In co-doped ZnO; AGZO = Al, Ga co-doped ZnO; TAZO = Sn, Al co-doped ZnO; ODL = optimal doping level (at%) of aluminum; T = transmittance (%) of AZO in the visible region; E_g_ = band gap (eV) of AZO; and *ρ* = resistivity (Ω cm) of AZO.

co-Doped ZnO	Process	Morphology	ODL (at%)	T (%)	E_g_ (eV)	*ρ* (Ω cm)	Ref.
AIZO	Sol-gel, ultrasonic spray pyrolysis	Grains	Al, In = 2	89.10	3.41	3.44 × 10^−3^	[71]
AIZO	Sol-gel, ultrasonic spray pyrolysis	Grains	Al, In = 1.5	79.62	3.53	2.74 × 10^−3^	[72]
AGZO	Sol-gel, spin coating	Micro-rods	Al, Ga = 1	95		23	[73]
AGZO	Aerosol-assisted CVD (AACVD)	Grains	-	~80	3.27–3.28	2.74 × 10^−2^	[25]
Sm:AZO	Nebulizer spray pyrolysis	Grains	Al = 3, Sm = 1	90	3.30	4.31 × 10^−4^	[12]
Pr:AZO	Nebulizer spray pyrolysis	Grains	Al = 3, Pr = 1.5	84–90	3.25	4.62 × 10^−4^	[11]
Ni:AZO	sol-gel, dip coating	Grains	-	-	~3.20–3.28	-	[76]
TAZO	Sol-gel, dip coating	Grains	Al, Sn = 1	88	~ 3.28	-	[74]
Ag:AZO	Sol-gel, spin-coating	Nanostructures	-	85	3.14	5.18 × 10^−2^	[75]

## Data Availability

The data used and/or analyzed in this mini review can be available upon reasonable requests.

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
