# Peer review of "Recent Progress in Solution Processed Aluminum and co-Doped ZnO for Transparent Conductive Oxide Applications"

_micromachines, 2023, doi:10.3390/mi14030536_

Round 1

Reviewer 1 Report

The paper of Mandeep Singh and Francesco Scotognella deals with solution processed ZnO doped with Al and Al co-doped. The topis is very wide, especially for AZO films which are believed to replace ITO in the terms of TCO applications. For me paper is written in chaotic way, there is no division into planar AZO films and its micro- and nanostructures what makes difficulties during the reading. There is also a lack of exemplary applications of AZO based layers as a TCO in optoelectronics devices.

1. the reference is a part of the sentence, therefore it has to be before the sentence ending sign (.) see lines 26, 30 ...

2. I noticed some missed space for instance lines 39, 40 before the referring to the bibliographic records

3. Copyrights issue of reprinted graphs

4. typo in line 138: Al3+

5. the review is based on papers published in last 7 years, why not 10 or 11?

6. For me there is no concise summary of the dopant concentration influence on optical and electrical properties of TCO layers

7. Paper with 56 references tends to be usual paper not a review

Author Response

The paper of Mandeep Singh and Francesco Scotognella deals with solution processed ZnO doped with Al and Al co-doped. The topis is very wide, especially for AZO films which are believed to replace ITO in the terms of TCO applications. For me paper is written in chaotic way, there is no division into planar AZO films and its micro- and nanostructures what makes difficulties during the reading. There is also a lack of exemplary applications of AZO based layers as a TCO in optoelectronics devices.

Response

We are thankful to the reviewer for commenting on our manuscript.

However, we are disagreed with the comment that the review is written in chaotic way. We do not think this type of division is required because at the end AZO should be transferred to substrate (most probably glass substrate) in the form of film for its application as a TCO. For example, in reference 22 (DOI: 10.1007/s11664-020-08107-9), the nanowires were deposited on FTO and quartz substrate. In this mini review, we have tried to include many different morphologies of AZO to presents the recent advance in synthesis of AZO and also to make the review interesting and valuable for readers.

For exemplary applications of AZO, we have added more references in the revised version of this mini review.

The point-to-point response of your each comment are as follows: 

Comment 1. the reference is a part of the sentence, therefore it has to be before the sentence ending sign (.) see lines 26, 30 ...

Response 1

We are thankful to the reviewer for pointing out this mistake. Now, throughout the revised manuscript changes have been made.

Comment 2. I noticed some missed space for instance lines 39, 40 before the referring to the bibliographic records

Response 2

Again, we are thankful to the reviewer for pointing out these typos. Missed spaces are now added throughout the revised manuscript.

Comment 3. Copyrights issue of reprinted graphs

Response 3

There is no copyright issue in the manuscript. In fact, before the submission we have taken all the permission of graphs or figures used in the manuscript. For your convenience, we are now sharing separate doc file in which we have listed permission of all the graphs/figures.

Comment 4. typo in line 138: Al3+

Response 4

Thanks for pointing out this typo. Correction has been made in the revised manuscript.

Comment 5. the review is based on papers published in last 7 years, why not 10 or 11?

Response 5

We have restricted ourselves to 7 years because of two reasons: 1. We want to presents the recent advance and 2. This is a mini review not comprehensive one.

Having said that, in the revised manuscript, we have added more discussion and references to enhance the length of review to 20 pages.  

Comment 6. For me there is no concise summary of the dopant concentration influence on optical and electrical properties of TCO layers

Response 6

According to your comment, we have restructured whole section to make it more concise and thoughtful in the revised manuscript. Changes are made in the revised mansucript.

Comment 7. Paper with 56 references tends to be usual paper not a review

Response 7

We are thankful to the reviewer for pointing this out. We have cited more references in revised manuscript.

Reviewer 2 Report

This paper is a mini-review about the state of the art of Al doped ZnO thin films growth by wet chemical methods. The main purpoise is the systematic review of the different works related to the topic in the last 7-years. The author have made special emphasis in the amount of dopant (or co-dopant) and how it affects the morphology, grain size, transmitance or conductivity for each process.

In the introduction, the importance of AZO as TCO is clearly described, as well as a general introduction in the synthesis of Al-doped ZnO. However, being a mini-review, no new-results are presented. Due to this is a literature review, no revisions related to experimental or analyisis are needed and I consider that the scheme and presentations of the information are correct.

Some typos should be corrected (lines 103, 109, 138, 171, 240, 242..)

Author Response

Comments and Suggestions for Authors

This paper is a mini-review about the state of the art of Al doped ZnO thin films growth by wet chemical methods. The main purpoise is the systematic review of the different works related to the topic in the last 7-years. The author have made special emphasis in the amount of dopant (or co-dopant) and how it affects the morphology, grain size, transmitance or conductivity for each process.

In the introduction, the importance of AZO as TCO is clearly described, as well as a general introduction in the synthesis of Al-doped ZnO. However, being a mini-review, no new-results are presented. Due to this is a literature review, no revisions related to experimental or analyisis are needed and I consider that the scheme and presentations of the information are correct.

Some typos should be corrected (lines 103, 109, 138, 171, 240, 242..)

Response

We are thankful to the reviewer for the comments, suggestions and recommending our mini-review for publication.

The typos pointed out by reviewer are now been corrected throughout the revised manuscript.

Reviewer 3 Report

According to manuscript micromachines-2160237, entitled „ Recent Progress in Solution Processed Aluminum and co-doped ZnO for Transparent Conductive Oxide Application“ by authors M. Singh and F. Scotognella.

 The paper presents a review on solution obtained ZnO, doped and co-doped with Al. The aim for studying ZnO:Al based materials is the replacement of ITO and development of efficient  transparent conducting oxides. The focus of the paper is the solution processing route for preparation of doped and co-doped ZnO structures.

The paper can be published after major corrections:

1. The text must be carefully checked. There are some unclear sentences. For example „Especially, in solution processing methods, the sol-101 ubility of precursors in solvent under considerations plays an extremely important role 102 and their always be a limit after which precursors didn’t dissolved.“ – page 3, rows 100 – 101.

2. Check English – „it greatly influence“ page 3, row 99; page 3 row 105 „this factors“; page 3, row 110 etc.

3. The authors state that „In fact, Al doping concentration was found to greatly affect 106 the structural, optical electrical properties of Al-doped ZnO thin films synthesized via sol-107 gel assisted spray-pyrolysis technique [41].“ Actually, doping concentration influences the properties of ZnO independent on the preparation method. It can be said, not „in fact“, but „for example“.

4. Page 3, rows 129 – 130 unclear text

5. The text from page 1 to page 3 describes ZnO:Al, but suddenly section 4.1. begins AZO review of the recent 7 years. The paper must be re-arranged. It can be divided in review of AZO and co-doped ZnO.

6. The title is a little misleading as „Aluminum and co-doped ZnO“ suggests review of different dopants, but in the presented manuscript, the authors describes only In and Al co-doped ZnO. There are many papers for Al, Ga co-doped ZnO; Al, N co-doped ZnO; Al, Cu co-doped ZnO; etc. The title can specified „Al and In co-doped ZnO“

7. Similar to Table 1, it will be usefull to add Table for Solution processed AL and In co-doped ZnO materials and their their functional properties

Author Response

Comments and Suggestions for Authors

According to manuscript micromachines-2160237, entitled „ Recent Progress in Solution Processed Aluminum and co-doped ZnO for Transparent Conductive Oxide Application“ by authors M. Singh and F. Scotognella.

 The paper presents a review on solution obtained ZnO, doped and co-doped with Al. The aim for studying ZnO:Al based materials is the replacement of ITO and development of efficient  transparent conducting oxides. The focus of the paper is the solution processing route for preparation of doped and co-doped ZnO structures. 

The paper can be published after major corrections:

 Response

We are thankful to the reviewer for commenting on our manuscript. The point-to-point response of your each comment are as follows: 

Comment 1. The text must be carefully checked. There are some unclear sentences. For example „Especially, in solution processing methods, the sol-101 ubility of precursors in solvent under considerations plays an extremely important role 102 and their always be a limit after which precursors didn’t dissolved.“ – page 3, rows 100 – 101.

Response 1

We are thankful to the reviewers for pointing out these mistakes. In the revised manuscript, we have not only modified this statement, but also made significant changes in the other parts of manuscript.

Comment 2. Check English – „it greatly influence“ page 3, row 99; page 3 row 105 „this factors“; page 3, row 110 etc. 

Response 2

We are again thankful to the reviewer of pointing these mistakes. We have checked thr English grammar, typo etc. throughout the revised manuscript and made significant changes. 

Comment 3. The authors state that „In fact, Al doping concentration was found to greatly affect 106 the structural, optical electrical properties of Al-doped ZnO thin films synthesized via sol-107 gel assisted spray-pyrolysis technique [41].“ Actually, doping concentration influences the properties of ZnO independent on the preparation method. It can be said, not „in fact“, but „for example“.

Response 3

We are apologizing to the reviewer for this confusing statement. By following your comment, we have re-structured whole section 3 to make it more concise and thoughtful.

Comments 4. Page 3, rows 129 – 130 unclear text

Response 4

Changes have been made in the revised manuscript.

Comment 5. The text from page 1 to page 3 describes ZnO:Al, but suddenly section 4.1. begins AZO review of the recent 7 years. The paper must be re-arranged. It can be divided in review of AZO and co-doped ZnO.

Response 5

We don’t know from where this confusion came, but starting from introduction we have clearly defined that this mini review is focused on AZO and co-doped ZnO. AZO is the abbreviation of aluminium doped ZnO that we have clearly defined both in abstract and introduction.

Comment 6. The title is a little misleading as „Aluminum and co-doped ZnO“ suggests review of different dopants, but in the presented manuscript, the authors describes only In and Al co-doped ZnO. There are many papers for Al, Ga co-doped ZnO; Al, N co-doped ZnO; Al, Cu co-doped ZnO; etc. The title can specified „Al and In co-doped ZnO“.

Response 6

We believe reviewer have skipped few parts of the review. Besides, Al, In co-doped ZnO, the reports on other co-doped ZnO such as (Pr, Al), (Sm, Al), (Ga, Al) co-doped ZnO are included in the mini-review. For your convenience, we have highlighted those parts in yellow in the revised manuscript. Moreover, now we have also included other co-doped ZnO. Changes are made in the revised manuscript.

Comment 7. Similar to Table 1, it will be usefull to add Table for Solution processed AL and In co-doped ZnO materials and their their functional properties.

Response 7

We are thankful to the reviewer for this comment. Table 2 for co-doped ZnO is now added in the revised manuscript.

Round 2

Reviewer 1 Report

I ma sorry to said but I am not convinced by authors about this paper, I am not satisfy with the corrections made by authors. See for instance:

- line 110:  Detailed theoretical observations reveal that the band gap and electrical conductivity of AZO were found to be decreased with increase in ‘x’ i.e. doping level of Al. - ZnO and ZnO:Al (AZO) have been successfully synthetized since 2016 when paper [54] was published so the question is why authors refer to theoretical paper instead of experimental one. I am not convinced with the statement that electrical conductivity of AZO were found to be decreased with increase in ‘x’ i.e. doping level of Al what is not true because Al doping of ZnO increase carrier concentration and therefore electrical conductivity

https://doi.org/10.1016/S0040-6090(99)00357-0

http://dx.doi.org/10.12693/APhysPolA.129.A-36

and many more

Moreover the statements form line 110 in in contrast with statement from line 449: The AZO film deposited via spray pyrolysis with  aluminium doping level of 2 at% was found to exhibit lowest resistivity i.e. 5.10×10 −3  Ωcm

- lines 113-116: he calculated band gap of pure ZnO  was found to consist of three parts: upper part (-3.8 to 0 eV), lower part (-6.7 to -3.8 eV)  and third part resulted from the interactions between lower and upper parts which is lied  between -18.6 to -16.0 eV. - what does it mean that band gap has negative value? In my opinion band gap is represented by one value, not two as authors claims. These information are ambiguous.

In my opinion presented work does not add anything new to the current state of knowledge

Reviewer 3 Report

The revised manuscript is improved and it can be published in its present form.